# Influence of Cooling Methods on the Residual Mechanical Behavior of Fire-Exposed Concrete: An Experimental Study

**DOI:** 10.3390/ma12213512

**Published:** 2019-10-26

**Authors:** Espedito Felipe Teixeira de Carvalho, João Trajano da Silva Neto, Paulo Roberto Ribeiro Soares Junior, Priscila de Souza Maciel, Helder Luis Fransozo, Augusto Cesar da Silva Bezerra, Antônio Maria Claret de Gouveia

**Affiliations:** 1Postgraduate Program in Materials Engineering, Federal University of Ouro Preto, Ouro Preto 35400-000, Brazil; espedito.carvalho@gmail.com (E.F.T.d.C.); fransozohelderluis@gmail.com (H.L.F.); amclaretgouveia@gmail.com (A.M.C.d.G.); 2Federal Institute of Education, Science and Technology of Minas Gerais, Ipatinga 35164-261, Brazil; joao.trajano@ifmg.edu.br; 3Department of Materials Engineering, Federal Centre of Technological Education of Minas Gerais, Belo Horizonte 30421-169, Brazil; 4Department of Transports Engineering, Federal Centre of Technological Education of Minas Gerais, Belo Horizonte 30421-169, Brazil; pmaciel@ymail.com

**Keywords:** concrete fire degradation, residual mechanical behavior, cooling method

## Abstract

This work reports the main conclusions of a study on the mechanical behavior of concrete under ISO 834 fire with different cooling methods. The aim of this research was to provide reliable data for the analysis of structures damaged by fire. The experimental program used cylindrical concrete test specimens subjected to ISO 834 heating in a furnace up to maximum gas temperatures of 400, 500, 600, 700, and 800 °C. The compressive strength was measured in three situations: (a) at the different temperature levels reached in the furnace; (b) after a natural cooling process; and (c) after aspersion with water at ambient temperature. The results indicate that the concrete residual compressive strength is fairly dependent on the maximum temperature reached in the furnace and revealed that concrete of a lower strength preserved relatively higher levels of strength. The cooling method significantly influenced the strength, albeit at a lower intensity. In all cases, the residual strength remained in the range of 38% to 67% of the strength at ambient temperature. The statistical analysis showed that the data obtained by the experimental program are significant and confirmed the influence of the conditions imposed on the residual strength.

## 1. Introduction

In recent decades, the study of concrete during and after a fire has been scant in Brazil. In the mid-1970s, after two buildings were damaged by fire, the development of fire safety technology became a challenge for engineers and fire authorities. A technical standard for the design of concrete structures under fire was published in 1976. It required fire resistance times for building structures, depending on their height and occupancy. This standard prescribed adequate concrete covering according to the required times for fire resistance. In general, considering low- and medium-rise buildings in Brazil (up to 15 floors), normal coverings of 25 mm were sufficient to achieve the desired fire resistance. In Europe and the United States, design guidelines are available in the Eurocodes (EN 1992-1-2) and American Standards (ACI/TMS 216.1-14). These standards emphasize the structural fire resistance of buildings and explicitly incorporate the verification of structural elements under fire conditions. A fire in a building can have huge consequences, putting human lives at risk. Nonetheless, concrete structures have a relatively high fire resistance and can often survive fire exposure. As such, it is relevant to investigate how to repair the damaged parts of a structure after a fire and thus avoid the costs of demolition and reconstruction. To estimate repair costs, it is necessary to assess the damage to a structure caused by fire [1].

Several authors have reported studies on the influence of a fire on concrete properties, with the residual compressive strength being one of the most important. There is a great variety of reported research considering the influence of parameters such as the water cement ratio, aggregate density, microsilica rate, heating process, state of initial stress, and state of stress during tests on the residual strength of concrete [2,3,4,5,6]. Malhotra [7] reported the main conclusions of early research up to 1982. Hertz [8], Riley [9], Saad et al. [10], Morsy et al. [11], and Neville [12] studied the effects of temperature on concrete, with a special focus on the effects on its microstructure. Scheinherrová et al. [13], Eidan et al. [14], Novak and Kohoutkova [15], and Abid et al. [16] evaluated the effect of fiber addition on concretes subjected to high temperatures. Agrawal and Kodur [17] investigated the residual performance of concrete beams after a fire in terms of the residual strength, structural response, and cracking pattern. Missemer et al. [18] evaluated the spalling of concrete when exposed to fire, from a global to microstructural analysis. Wu et al. [19] studied the residual mechanical properties of concrete containing aggregates from demolition material. Fernandes et al. [20] reviewed the state of the art of concretes subjected to high temperatures, focusing on the microstructure of the material, physicochemical changes, and analysis techniques. Rajawat et al. [21] studied the performance of concrete containing fine white ceramic aggregates. Dauti et al. [22] analysed the percolation of moisture in concrete at elevated temperatures using the technique of neutron tomography.

Xotta et al. [23], Sinaie et al. [24], and Hertz [25] sought to develop a behavior model of concrete properties when exposed to high temperatures. Considering only the residual compression strength of concrete, as a general conclusion, it was found that it decreases with an increasing temperature by approximately 40% to 50% of the ambient temperature strength, with the temperatures in the range of 450 to 550 °C being crucial in this phenomenon. During heating, surface cracks are visible at 600 °C, becoming more pronounced at 800 °C [2,17].

Therefore, the effect of high temperatures on the strength of normal concrete has been extensively investigated. Nevertheless, it is important to mention that the influence of the cooling process after high temperature exposure has been much less investigated, although the compressive strength is significantly influenced by the cooling method. Depending on the cooling method, an additional strength reduction up to 38% occurs for water-cooled specimens compared to slowly cooled specimens. However, the difference between cooling methods decreases when the concrete is exposed to higher temperatures. This is due to the bond strength between concrete and the reinforcement, which is very sensitive to high temperatures and subsequent cooling. A gradual decrease of the bond strength is observed until 600 °C [1]. Bingöl and Gül [26] performed similar evaluations and studied the effect of the cooling regime on two concrete classes with initial strengths of 20 and 35 MPa. Specimen heating was performed up to 700 °C, and cooling occurred using two methods: rapid cooling in water and air. Up to 400 °C, air-cooled concrete maintained 80% of its original strength, while an average loss of 30% strength was observed in water-cooled specimens. After 400 °C, both types of concrete lost their strength rapidly, and the strength loss was greater in water-cooled specimens. Strength loss was more significant for the specimens rapidly cooled in water. Both concrete mixtures lost a significant part of their initial strength when the temperature reached 700 °C. Zhai et al. [27] studied the influence of heating cycles at high temperatures and subsequent water-cooling on the mechanical properties of concrete, using impact tests. Zhang, Yuan, and Dong [28] concluded that the cooling method is most influential between 200 and 500 °C, regardless of the type of aggregate used. Chan, Luo, and Sun [29] realized that the type of cooling has an impact on the residual porosity and consequently affects strength. Tanaçan, Ersoy, and Arpacıoglu [30] investigated aerated concrete and concluded that an abrupt water cooling regime soon after heating increases cracking, with a subsequent strength drop. Lee, Xi, and Willian [31] found that the stiffness, strength, and permeability are influenced by heating and cooling cycles. Karakoç [32] evaluated the compressive strength of light aggregate concretes and demonstrate that it is influenced by the cooling methods. Kim et al. [33] concluded that the cooling rate affects the dimensional variation of the aggregates and that the mechanical properties are deteriorated by accelerated cooling. Xiang et al. [34] realized that, for slow cooling, the heating temperature influences the residual cracking pattern and failure mode during compression testing.

In this context, it is essential to understand the behavior of concrete structures affected by fire and the influence of heat on building materials. Several studies have evaluated the residual properties of cement-based materials after a high temperature, whilst others have investigated the physical and chemical changes occurring at the microstructural level. However, comprehensive studies evaluating the influence of cooling methods on the mechanical properties of concrete are scarce in the literature. Therefore, the research reported here aims to broaden the knowledge on the subject, providing new results, and was designed to answer two questions: How is the residual strength of concrete under ISO 834 fires affected by the strength at ambient temperatures? To what extent does the cooling of hot concrete specimens affect the residual strength? These questions are particularly important for engineers and fire brigade officials when examining fire-damaged concrete structures.

## 2. Experimental Program

### 2.1. Materials and Mix Proportion

The coarse aggregates were limestone with a maximum diameter of 25 mm and fineness modulus of 7.05. A sand of fineness modulus 2.68 and a cement ASTM Type III [35] with a compression strength of 38.1 MPa were used. The dosage process considered theoretical compressive strengths (f_ck_) of 15, 21, 25, and 35 MPa with water/cement ratios of 0.620, 0.517, 0.458, and 0.338, respectively. To determine the optimal ratio between sand and gravel, the modified Reilly [36] method as exposed in Carvalho [37] was used. The result was a minimum porosity with a mixture of 65.5% coarse aggregates and 34.5% sand. The water/dry materials ratio (A%) was related to the fineness modulus using 11 distinct dosages whose slump tests were in the range of 55 to 92 mm. The water/cement ratio was determined for concrete of slump 70 mm. The A_G_ characteristic of aggregates following the Reilly [36] method was 0.474. The Reilly constants were M_1_ = 4.2031 and M_2_ = 0.3281. All specimens were kept at ambient humidity for 90 days until testing.

### 2.2. Specimen Preparation and Curing

The experimental program included 252 cylindrical concrete specimens that were 100 mm in diameter and 200 mm in height. Initially, the materials were separated and weighed according to the predefined proportions. Then, the components were mechanically mixed in the following order: coarse aggregates, sand, half of the water content, Portland cement, and the remaining water. After obtaining a homogeneous mixture, the fresh concrete was poured into the cylindrical molds. Manual densification was performed to reduce voids and air bubbles, ensuring a uniform mass. After 24 h of molding, the stiffened specimens were removed from the molds and allowed to wet cure for 28 days, which was a sufficient time for cement hydration reactions and strength gain.

### 2.3. Heating and Cooling Methods

The concrete test specimens were subjected to ISO 834 heating in a cylindrical furnace of a 770 mm diameter and 800 mm height, up to maximum gas temperatures (θ_gmax_) of 400, 500, 600, 700, and 800 °C (48 specimens at each temperature). A typical time–temperature curve of the furnace is shown in Figure 1. For each temperature in the furnace, three situations have been considered: hot tested (HT), naturally cooled (NC), and cooled after spraying with room temperature water (WAC). As for the HT situation, the model is rated still hot according to the heating temperature. In the second methodology, the material is naturally cooled in air for 24 h. Finally, in WAC (Figure 2), the cooling is done abruptly, simulating a fire situation where water is applied to the structure at a high temperature, in which there is a great thermal shock in the material.

### 2.4. Test Set-Up

The strength was evaluated by the compression test, with a slow load application, to configure a quasi-static condition. The tests were performed on universal servo hydraulic testing equipment, permitting accurate load measurements to be obtained. All conditions were analysed (theoretical strength, temperature, and cooling method). The control group was tested at room temperature for each theoretical strength.

### 2.5. Statistical Approach

The consistency of the data obtained was evaluated by statistical analysis. The significance level adopted was 5% (0.05). Therefore, the analyses had a confidence interval of 95%. Univariate analysis of variance (ANOVA) using the F-ratio was chosen as the analysis technique. The impact of the conditions imposed on the compressive strength was measured by the size of the effect, using the variances and the parameter eta-square (η^2^). The null effect was identified by the value zero, and values close to or greater than 1 indicated a broad effect. The observed power represented the robustness of the tests in identifying the differences between the groups analysed. Values close to one indicated a high test power.

## 3. Results and Discussion

### 3.1. Mechanical Behavior

Table 1, Table 2 and Table 3 summarize all the residual compressive strength results. The main values represent the average of four specimens. The percentage of maintenance of strength as a function of the result without heating is shown within brackets, and the respective standard deviation is expressed in parentheses. The data served as the basis for the statistical analysis and for the evaluation of the mechanical behavior as a function of the imposed conditions. Figure 3, Figure 4 and Figure 5 show the behavior of the concrete compressive strength of 15, 21, 25, and 35 MPa for each temperature in the furnace considering the test conditions. Each point represents the average of four specimens. The lines that connect the experimental means show the tendency of residual strength.

In the first analysis, the strength had a similar trend. There was a decrease in strength as the temperature increased, especially between room temperature and 400 °C. This result seems to be reasonable considering that 400 °C was reached in all experiments, which is the temperature which corresponds to major damage to the concrete microstructure, as observed by several authors [10,11,12]. Between 400 and 600 °C, the loss of strength was subtle, yet significant. After 600 °C, there was a sudden reduction in the strength capacity. Some tests resulted in strength gain; however, there were specific occurrences, most likely due to the variability of the data (Figure 3, f_ck_ = 25 MPa, 500 °C and Figure 5, f_ck_ = 35 MPa, 500 °C). For heating up to 800 °C, the residual compressive strength remained in the range of 38% to 67% of the ambient temperature strength. Natural air cooling resulted in lower decreases in strength, while the greatest variations were obtained with water spray. The hot test resulted in an intermediate condition among the others. Similar results were obtained by Chan, Luo, and Sun [29]. They concluded that abrupt cooling by immersion in water promotes greater decreases in compressive strength compared to gradual cooling inside the furnace. Furthermore, they observed that higher temperatures imply a higher porosity and consequently lower strength. The lowest relative residual strengths (%) were reached by the concretes with the highest theoretical strengths. As also verified by Botte and Caspeele [1], there was a tendency in which the difference between both cooling methods decreased when the concrete was exposed to higher temperatures. 

### 3.2. Statistical Analysis

Table 4 shows the ANOVA results. The corrected model and the intercept, the standard models of the analysis program, were significant (*p* < 0.05), which enabled the variables to be correlated analytically. The categorical variable test method (M), temperature (T), and theoretical strength (f_ck_) were significant (*p* < 0.05), which indicates a relevant effect on the compressive strength. The effect size was moderate for the test method and more intense for the temperature and strength variables. However, the values obtained indicate a significant influence of the conditions on the strength. The power observed was the maximum for all situations evaluated. The adopted model can explain 88% of the observed values (coefficient of determination R_square_ = 0.880). Therefore, the obtained data are statistically significant and adequately explain the proposed problem.

### 3.3. Fire Degradation and Cooling Implications

Figure 6 identifies specimens in three different situations: (a) inside the oven during heating; (b) after slow cooling in the air; and (c) with and without the spalling phenomenon. The first notable point refers to the physical phenomena related to dimensional variation and cracking [3,15]. The specimens exposed to the flow of heated gases presented extensive cracking, both on the upper face (Figure 6b) and on the lateral ones (detail I of Figure 6c). The cracks were mainly due to volumetric dimensional variation due to the different temperature gradients. Furthermore, the surface exposed to the environment dissipated heat faster than the interior of the material, leading to different dimensional variations along the volume [38]. Cementitious materials, included among ceramic materials, are fragile, they do not tolerate strains, and their performance is significantly influenced by internal imperfections (pores, cracks, and defects in the microstructural arrangement) [39,40,41]. Concrete, idealized as a particulate composite, has constituents (cementitious matrix, sand, and gravel) that have different thermal expansion coefficients [42]. Furthermore, the interfacial transition zone (ITZ) functions as a site capable of promoting crack propagation [43,44]. The distinct dimensional variation between the constituents significantly affects the ITZ, promotes cracking of the material, increases defects, and consequently reduces the strength. The internal stresses acting during loading are distributed in the material volume and are amplified at the ends of the cracks [45]. At the maximum loading stage, the material collapses and fails due to the unstable mechanism of crack propagation [46,47]. Another point concerns the degradation of the cement matrix and the particulate constituents, which deteriorate with increasing heat. Phase changes, chemical changes, the evaporation of free water, and the release of chemically bound water occur, which affect the strength drops [48]. The phenomenon of fragmentation occurred in some specimens (detail II of Figure 6c), with considerable material loss. Fragmentation results from the pressure exerted by water vapor in the capillary pores and the internal stresses generated by the thermal gradient [18].

If there is no material fragmentation during heating, chemical and physical phenomena, acting in a synergistic manner, promote a decrease in strength. Under the three conditions evaluated, the mechanisms of chemical and structural degradation mostly occurred during heating. Therefore, the portion of lost strength related to cooling methods is governed by physical phenomena [4]. Abrupt cooling is the most severe condition. When the material encounters water, it experiences a marked temperature difference; thus, the heat exchanges intensify, as do the dimensional variations. Both the cementitious matrix and the aggregates are damaged; however, the ITZ is highly damaged (defects and microcracking). The result is a marked decrease in strength compared to other conditions. Slow cooling is the best condition, even with a decrease in strength. By losing energy slowly, the material accommodates the dimensional variations more efficiently, resulting in less damaged constituents and ITZs [14,15,16]. The hot test is an unfavourable thermodynamic condition due to the marked energy level of the system, thus resulting in an intermediate condition between those evaluated. The decrease in residual strength that occurs as the theoretical strength increases can be explained in a similar way. Generally, a higher strength implies a denser, more cohesive cementitious matrix with a lower porosity. However, the material ability to accommodate stresses and strains decreases, i.e., the material becomes less tolerant to strains. Therefore, considering the heating and cooling stages, the greater the strength, the greater the damage to the material due to the dimensional variation, which results in more severe cracking, degradation, and irregularities in the ITZ [27].

## 4. Conclusions

This work presents a case study on the effect of the cooling method on the mechanical behavior of concrete after a fire according to ISO 843. Based on the results of the experimental study, the following conclusions can be formulated:(1)The combined conditions (temperature, theoretical strength, and cooling method) had a significant influence on the residual strength. In general, the worst-case scenario was obtained considering high temperatures in the order of 800 °C, sudden cooling with water spray, and higher theoretical strengths. The best-case scenario was obtained with a 400 °C temperature, slow air cooling, and the lowest theoretical strengths;(2)Sudden cooling with water spraying was the most severe condition, resulting in the greatest decreases in strength. This result can be explained by the sudden dimensional variation of the particulate constituents and the cementitious matrix, which generates cracks and weakens the ITZ;(3)Concretes with a higher theoretical strength showed a lower residual strength. A higher strength implied a denser cementitious matrix, a lower porosity, and more consistent ITZ. However, the accommodation of stresses and strains was impaired, resulting in more extensive cracking and more severe degradation of the material;(4)The higher the temperature, the greater the losses of strength due to the degradation of concrete by physical and chemical phenomena. During heating, there was evaporation of free water, the release of chemically bound water, phase changes, and chemical changes;(5)Statistical analysis showed that the values obtained by the mechanical tests were significant and that the conditions evaluated (temperature, theoretical strength, and cooling method) significantly influenced the residual strength;(6)In future studies, it is suggested that changes in the structure of the concrete cement matrix are studied using a scanning electron microscope (SEM) to aggregate information about changes in the matrix as a function of heating and cooling processes.

## Figures and Tables

**Figure 1 materials-12-03512-f001:**
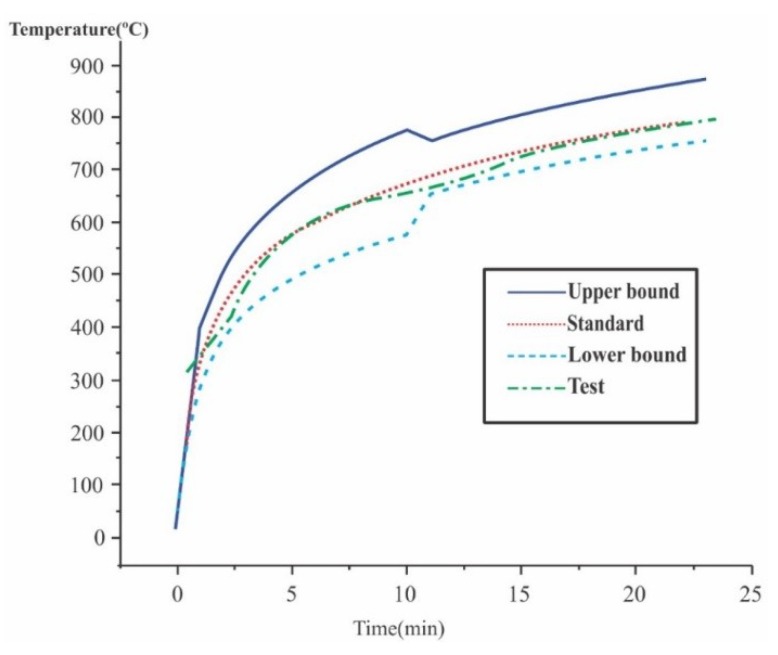
ISO 834 heating curve and furnace heating regime.

**Figure 2 materials-12-03512-f002:**
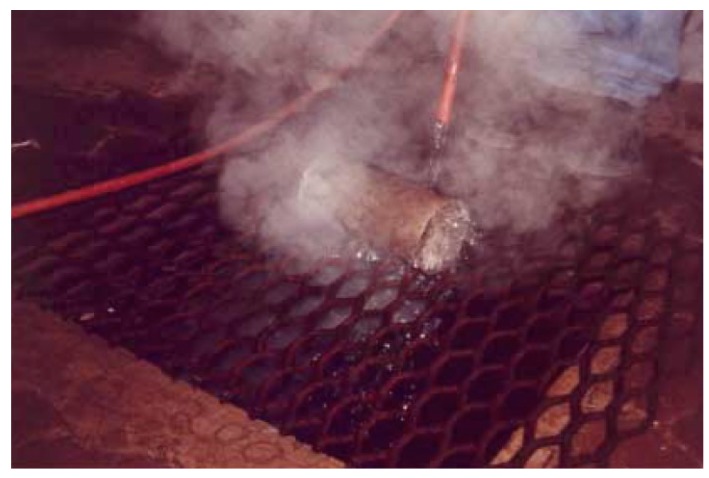
Subtle cooling of test specimens by water aspersion.

**Figure 3 materials-12-03512-f003:**
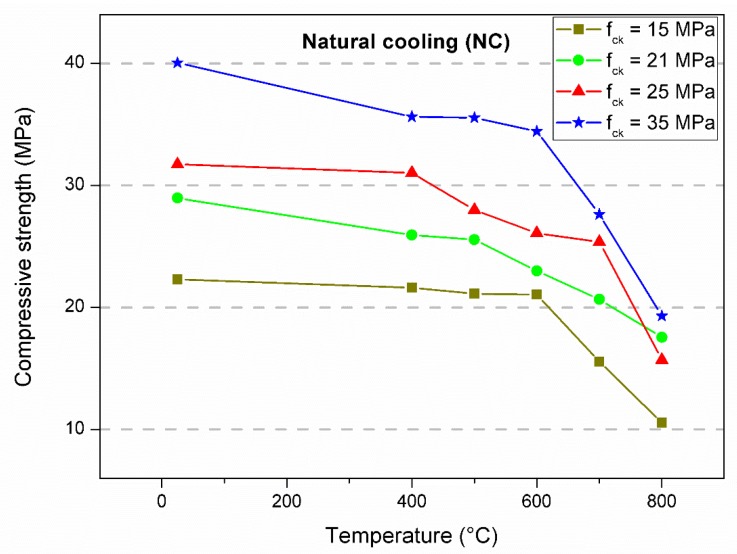
Compressive strength average of specimens cooled naturally.

**Figure 4 materials-12-03512-f004:**
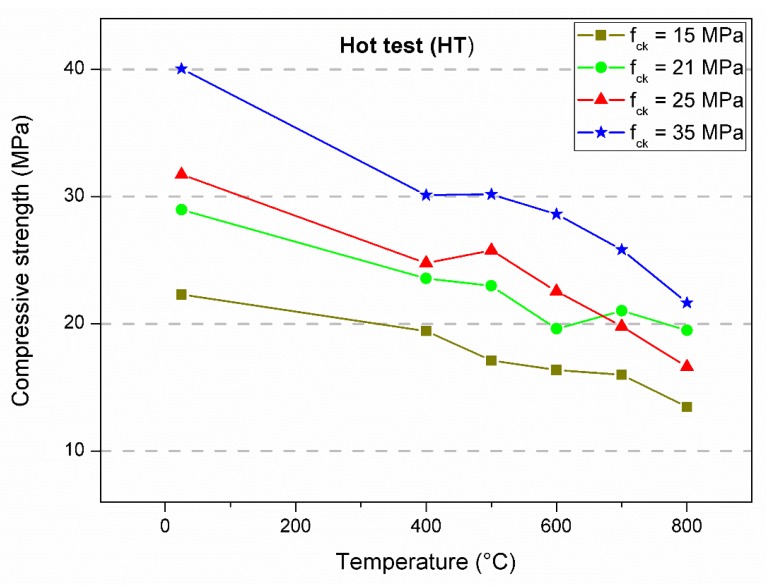
Compressive strength average of specimens tested when hot.

**Figure 5 materials-12-03512-f005:**
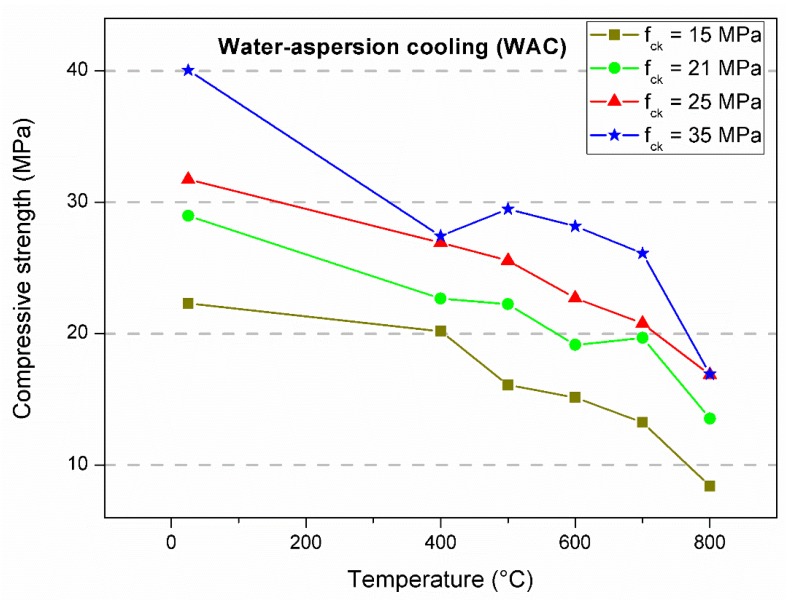
Compressive strength average of specimens cooled after aspersion with water at ambient temperature.

**Figure 6 materials-12-03512-f006:**
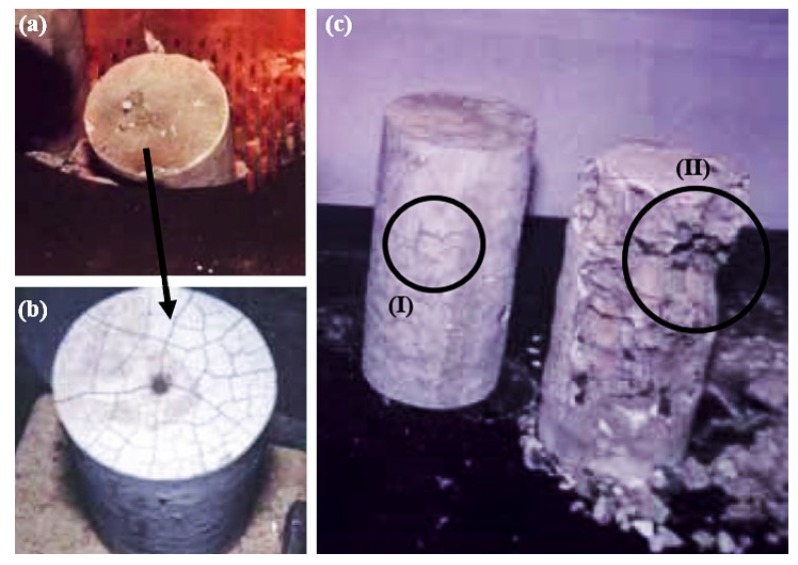
Test specimens (**a**) during heating inside the furnace and (**b**) after slow air cooling, and (**c**) evidence of the fragmentation phenomenon. (**I**) extensive cracking and (**II**) phenomenon of fragmentation.

**Table 1 materials-12-03512-t001:** Results of the compressive strength for natural cooling.

Theoretical Strength ^1^ (f_ck_)	Reference 20 °C	Cooled Naturally (NC)				
		400 °C	500 °C	600 °C	700 °C	800 °C
15	22.29 ^1^ [100] ^2^	23.02 [103]	22.54 [101]	21.04 [94]	15.55 [70]	10.55 [47]
	(0.43) ^3^	(2.54)	(2.25)	(3.35)	(1.52)	(1.66)
21	28.94 [100]	25.91 [89]	25.53 [88]	22.98 [79]	20.64 [71]	17.54 [61]
	(0.93)	(3.35)	(2.03)	(2.03)	(2.87)	(2.82)
25	31.71 [100]	30.99 [98]	27.97 [88]	26.07 [82]	25.35 [80]	15.69 [50]
	(1.28)	(2.91)	(1.18)	(2.32)	(3.25)	(1.75)
35	40.02 [100]	35.59 [89]	35.52 [89]	34.40 [86]	27.59 [69]	19.29 [48]
	(2.23)	(5.08)	(5.71)	(2.39)	(4.47)	(2.97)

Notes: ^1^ compressive strength in MPa; ^2^ percentage of maintenance of strength; ^3^ standard deviation.

**Table 2 materials-12-03512-t002:** Results of the compressive strength for hot tests.

Theoretical Strength ^1^ (f_ck_)	Reference 20 °C	Hot Test (HT)				
		400 °C	500 °C	600 °C	700 °C	800 °C
15	22.29 ^1^ [100] ^2^	19.42 [87]	17.11 [77]	16.35 [73]	15.98 [72]	13.46 [60]
	(0.43) ^3^	(2.55)	(1.49)	(1.96)	(2.17)	(1.55)
21	28.94 [100]	23.54 [81]	22.98 [79]	19.61 [68]	21.01 [73]	19.48 [67]
	(0.93)	(2.71)	(1.06)	(2.78)	(2.65)	(1.37)
25	31.71 [100]	24.75 [78]	25.75 [81]	22.54 [71]	19.79 [62]	16.61 [52]
	(1.28)	(3.52)	(0.63)	(2.16)	(0.97)	(2.49)
35	40.02 [100]	30.09 [75]	30.16 [75]	28.59 [71]	25.82 [65]	21.63 [54]
	(2.23)	(3.63)	(2.46)	(4.03)	(2.64)	(3.18)

Notes: ^1^ compressive strength in MPa; ^2^ percentage of maintenance of strength; ^3^ standard deviation.

**Table 3 materials-12-03512-t003:** Results of the compressive strength for water aspersion cooling.

Theoretical Strength ^1^ (f_ck_)	Reference 20 °C	Cooled after Water Aspersion (WAC)				
		400 °C	500 °C	600 °C	700 °C	800 °C
15	22.29 ^1^ [100] ^2^	20.17 [90]	16.08 [72]	15.14 [68]	13.24 [59]	8.41 [38]
	(0.43) ^3^	(1.74)	(2.35)	(1.24)	(1.66)	(1.07)
21	28.94 [100]	22.67 [78]	22.23 [77]	19.14 [66]	19.67 [68]	13.52 [47]
	(0.93)	(2.89)	(3.52)	(2.08)	(2.47)	(1.35)
25	31.71 [100]	27.06 [85]	25.54 [81]	22.69 [72]	20.76 [65]	16.85 [53]
	(1.28)	(3.28)	(2.30)	(2.85)	(1.66)	(0.94)
35	40.02 [100]	27.41 [68]	29.47 [74]	28.15 [70]	26.09 [65]	16.92 [42]
	(2.23)	(2.86)	(2.96)	(3.46)	(3.64)	(3.61)

Notes: ^1^ compressive strength in MPa; ^2^ percentage of maintenance of strength; ^3^ standard deviation.

**Table 4 materials-12-03512-t004:** Results of the statistical analysis.

Effect	*p*-Value	Effect Size	Observed Power
Corrected Model	0.000	0.880	1.000
Intercept	0.000	0.989	1.000
Test method (M)	0.000	0.285	1.000
Temperature (T)	0.000	0.689	1.000
Strength (f_ck_)	0.000	0.760	1.000

Notes: The underlined values are significant; R_square_ = 0.880.

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
