# Peer review of "Influence of Cooling Methods on the Residual Mechanical Behavior of Fire-Exposed Concrete: An Experimental Study"

_materials, 2019, doi:10.3390/ma12213512_

Round 1
Reviewer 1 Report
The manuscript is well prepared, written, and organized. This paper has the technical merit necessary to benefit concrete materials engineers/experts as well as the journal readers. Furthermore, there are limited articles on the three methods studied in the manuscript.
The Results and Discussion states:
"Figures 3-5 show the behaviour of the concrete compressive strength of 15, 21, 25 and 35 MPa for each temperature in the furnace considering three
situations: hot tested (HT), naturally cooled (NC) and cooled after spraying with room temperature water (WAC)"
The reviewer suggests:
The authors should provide detailed procedures on the above three situations in the "Experimental Programme" section. I understand that there is a brief description; however, I strongly believe adding detailed procedures will benefit the readers.
Otherwise, the reviewer commends the authors for an excellent study.
Reviewer 2 Report
Please find attached a PDF file with my comments and suggestions for authors.

Reviewer 3 Report
The authors present a work on the Influence of Cooling Method on Residual Mechanical Behaviour of Fire-Exposed Concrete: Experimental Study. The subject of the authors work is an important significant issue in structural engineering and materials, and such an attempt is of great interest.
At the beginning of the article, the authors reviewed the literature on the subject of research. It is worth noting that many literature items are very new. However, in my opinion, this review could be broader and contain more details from the studies of other authors. The scope and description of the research presented below is comprehensible and factually correct. The analysis of the results obtained is also described correctly. It would also be advisable for the authors to study the changes in the cement matrix structure of concrete in the future using an optical method, e.g. SEM.
However the paper, in its present form, requires some substantial modifications in order to justify its publication in an International Journal such as Materials. I think that the following points should be further elaborated by the authors:
photographs shown in Figure 6 are blurred. Maybe you can give better quality, if possible, please extend the study part (provide more detailed information from the results of other researchers cited in this article).
I have no substantive comments.
Round 2
Reviewer 2 Report
My comments and suggestions have been addressed. The manuscript has been improved during the review process. I recommend to accept the manuscript for publication.